# Ciprofloxacin-Loaded Zein/Hyaluronic Acid Nanoparticles for Ocular Mucosa Delivery

**DOI:** 10.3390/pharmaceutics14081557

**Published:** 2022-07-27

**Authors:** Telma A. Jacinto, Breno Oliveira, Sónia P. Miguel, Maximiano P. Ribeiro, Paula Coutinho

**Affiliations:** 1CPIRN-UDI/IPG, Centro de Potencial e Inovação em Recursos Naturais, Unidade de Investigação para o Desenvolvimento do Interior do Instituto Politécnico da Guarda, Avenida Dr. Francisco de Sá Carneiro, No. 50, 6300-559 Guarda, Portugal; telmajacinto@ipg.pt (T.A.J.); brennofcb@hotmail.com (B.O.); spmiguel@ipg.pt (S.P.M.); mribeiro@ipg.pt (M.P.R.); 2CICS-UBI, Centro de Investigação em Ciências da Saúde, Universidade da Beira Interior, Avenida Infante D. Henrique, 6200-506 Covilhã, Portugal

**Keywords:** flash nanoprecipitation, conjunctivitis, nanoparticles, zein, hyaluronic acid, ciprofloxacin

## Abstract

Bacterial conjunctivitis is a worldwide problem that, if untreated, can lead to severe complications, such as visual impairment and blindness. Topical administration of ciprofloxacin is one of the most common treatments for this infection; however, topical therapeutic delivery to the eye is quite challenging. To tackle this, nanomedicine presents several advantages compared to conventional ophthalmic dosage forms. Herein, the flash nanoprecipitation technique was applied to produce zein and hyaluronic acid nanoparticles loaded with ciprofloxacin (ZeinCPX_HA NPs). ZeinCPX_HA NPs exhibited a hydrodynamic diameter of <200 nm and polydispersity index of <0.3, suitable for ocular drug delivery. In addition, the freeze-drying of the nanoparticles was achieved by using mannitol as a cryoprotectant, allowing their resuspension in water without modifying the physicochemical properties. Moreover, the biocompatibility of nanoparticles was confirmed by in vitro assays. Furthermore, a high encapsulation efficiency was achieved, and a release profile with an initial burst was followed by a prolonged release of ciprofloxacin up to 24 h. Overall, the obtained results suggest ZeinCPX_HA NPs as an alternative to the common topical dosage forms available on the market to treat conjunctivitis.

## 1. Introduction

Conjunctivitis affects many people worldwide and consists of the inflammation and swelling of the conjunctival tissue, as well as dilation of the blood vessels, ocular discharge, and discomfort. Conjunctivitis can be divided into four main groups based on the etiology: bacterial, viral, allergic, and irritant [1,2]. Bacterial conjunctivitis is the second-most common infectious conjunctivitis and is more frequent in children [3]. Further, bacterial conjunctivitis is one of the most common ophthalmic diseases in developed countries [4]. Several bacterial are etiological agents of conjunctivitis, the most common of which being *Streptococcus pneumoniae*, *Haemophilus influenza*, *Moraxella catarrhalis*, and *Staphylococcus aureus*; the last is more common in adults [5]. Ciprofloxacin (CPX) is one of the most-used antibiotics in the treatment of bacterial conjunctivitis [6], and is a broad-spectrum antibiotic that belongs to a class of antibiotics designed by fluoroquinolones [7]. Fluoroquinolones have excellent antibacterial effects against Gram-negative and many Gram-positive bacteria [8]. Nevertheless, commercial CPX eye drop solutions have an acidic pH, which causes local burning and itching [9,10]. Furthermore, the low solubility of CPX under ocular physiological conditions (pH ≈ 7) leads to a lower drug bioavailability [9].

Up to the present, topical dosage forms were elected as a less invasive administration route to the ocular mucosa. However, obstacles such as tear fluid production and the corneal barrier limit drug bioavailability. To tackle this challenge, researchers have proposed nanoparticles (NPs)-based systems for treating ocular infections, as reviewed by Liu et al. [11]. These systems can increase the retention time of the drugs on the ocular surface and protect them from enzymatic degradation, while simultaneously contributing to the decrease of the drug concentration administered to assure the therapeutic effect [12,13,14]. In addition, other authors demonstrated that by incorporating drugs into NPs, there is an enhancement of corneal permeability [11,15]. Hence, flash nanoprecipitation (FNP) is a simple and effective approach to producing NPs with high drug-encapsulation efficiency (EE) [16]. The FNP technique is based on a rapid mixing that creates high-supersaturation conditions, leading to the precipitation and encapsulation of both hydrophobic and hydrophilic drugs into polymeric NPs [16,17,18]. Several studies demonstrated that FNP allows the encapsulation of hydrophobic and hydrophilic drugs with high EE, as we have previously reported [18,19].

Herein, ZeinCPX_HA NPs were prepared using the FNP technique. Zein is a water-insoluble protein extracted from corn and is generally recognized as safe (GRAS) by the FDA [20]. In fact, zein has been widely explored in biomedical applications, namely in the field of pharmaceuticals, due to its physicochemical and biological properties [21]. Further, zein has been widely applied as a drug carrier due to its biocompatibility and amphiphilic nature which promote the self-assembly process [18,22,23,24] and the encapsulation of poorly water-soluble compounds [25]. On the other hand, hyaluronic acid (HA) is a polysaccharide selected due to its mucoadhesive character so that it will increase the pre-corneal residence of the drug [26]. Therefore, the pre-corneal clearance will be reduced by using HA, and consequently, a higher cellular interaction and ocular bioavailability will be attained [27]. Moreover, HA is a ligand for the CD44 receptor, which is present in the human cornea and conjunctiva. Under some pathological and inflammatory conditions, the CD44 receptor number increases, prompting the interaction with HA [28,29,30,31].

To the best of our knowledge, this is the first report of ZeinCPX_HA NPs produced by FNP. In the first instance, the polymer ratio (zein and HA) and cryoprotectants (glucose and mannitol) were optimized to obtain stable lyophilized NPs suitable for ocular drug delivery. These cryoprotectants can act as protective agents during freezing due to an increase in the surface tension of the water molecules, and can also work as cryoprotectants by preventing stress during the drying phase [32,33].

Thus, the main purpose of this study was to develop biocompatible polymeric NPs suitable for CPX ocular delivery with enhanced bioavailability and stability under long-term storage.

## 2. Materials and Methods

### 2.1. Materials

Dulbecco’s Modified Eagle Medium/Nutrient Mixture F-12 (DMEM-F12), phosphate-buffered saline (PBS) solution, 3-(4,5-Dimethylthiazol-2-yl)-2,5-diphenyltetrazolium bromide (MTT), trypsin, D(-)-Glucose, and high-performance liquid chromatography (HPLC) grade ciprofloxacin 98% were acquired from Sigma-Aldrich (Lisboa, Portugal). Ethanol (99.9%), glacial acetic acid, D(-)-Mannitol, dimethyl sulfoxide (DMSO) ≥ 99.9%, and HPLC-gradient grade acetonitrile were purchased from VWR Chemicals (Radnor, PA, USA). Normal human dermal fibroblasts (NHDFs) were purchased from PromoCell (Labclinics, S.A., Barcelona, Spain). Hydrochloric acid (HCl) was bought from Panreac (Barcelona, Spain). HPLC-grade ortho-phosphoric acid was acquired from Fisher Scientific (Oeiras, Portugal). Hyaluronic acid (MW: 1.0–2.0 Million Da) was purchased from Carbosynth (Bershire, UK). Zein (purified) was obtained from Acros Organics (Waltham, MA, USA). Fetal bovine serum (FBS) was obtained from Biowest (Riverside, MO, USA). Cell culture T-flasks were supplied by Orange Scientific (Braine-l’Alleud, Belgium). Ultrapure water was obtained by using a Q-POD^®^ dispenser (Merck Millipore, Burlington, MA, USA) (0.22 µm filtered; 18.2 MΩ/cm at 25 °C). Sodium bicarbonate ≥ 99% and ethanol were acquired from José Manuel Gomes dos Santos, LDA (Odivelas, Portugal). Sodium chloride 99.5% and potassium chloride ≥ 99.5% were purchased from Honeywell Fluka (Charlotte, NC, USA).

### 2.2. Production of the Zein and HA-Based NPs

The production of the Zein_HA NPs was achieved by the FNP technique using a confined impinging jets mixer (CIJM), which was produced at Fablab-IPG according to the model described by Han et al. [34]. Zein and HA (with pH ≈ 7) solutions were prepared, under agitation, in ethanol (80% *v*/*v*) and ultrapure water, respectively. Before mixing, the concentration of the zein solution was set at 2.5 and 5 mg/mL and the HA solution was set to 1 and 2.5 mg/mL. Briefly, the solution of zein (2.5 mL) was mixed against an equal volume of the HA solution. The NPs were collected in a solution of 45 mL of ultrapure water (pH ≈ 11).

### 2.3. Screening of Cryoprotectants

It is well documented that the long-term storage of the NPs is preferable under dried solid forms to liquid forms [35]. This way, glucose and mannitol (5 and 10% *w*/*v*) were added to the NPs solution before the freeze-drying process. Briefly, 1 mL of NPs solution with or without cryoprotectant (in a 1:1 ratio) were placed into glass vials and frozen at −80 °C for 12 h. Then the samples were lyophilized for 24 h in a Telstar LyoQuest −85 °C (Telstar, Madrid, Spain) operating a condenser at −67 °C and pressures below 0.05 mbar and lyophilized. After lyophilization, NPs were rehydrated by adding ultrapure water and left under static conditions for 10 min at room temperature before being fully reconstituted by manual shaking.

To evaluate the protective effect exerted by the cryoprotectants (5 and 10% *w*/*v*), the redispersibility index (RDI) was calculated according to the following Equation (1) [36]:(1)$$\text{RDI (\%)}=\frac{D}{D_{0}}\times 100$$
where D is the rehydrated NPs hydrodynamic diameter (Dh), and D_0_ is the Dh of fresh NPs. RDI values close or equal to 100% mean that samples can be appropriately resuspended.

### 2.4. Characterization of the NPs

#### 2.4.1. Particle Dh, Zeta Potential and Polydispersity Index (PDI) Measurements

Zein_HA NPs’ Dh, zeta potential, and PDI were determined by dynamic light scattering (DLS) using Zetasizer Nano ZS (Malvern Instruments, Malvern, UK). These parameters were also assessed for fresh and rehydrated NPs at room temperature.

#### 2.4.2. Fourier Transform Infrared (FTIR) Analysis

The FTIR spectra of the lyophilized NPs, drug, polymers, and cryoprotectant were acquired on a Nicolet iS10 spectrometer, with a 4 cm^−1^ spectral resolution from 500 to 4000 cm^−1^ and 128 scans per run (Thermo Scientific Inc., Waltham, MA, USA).

#### 2.4.3. Thermogravimetric (TGA) and Differential Scanning Calorimetry (DSC) Analysis

TGA and DSC analyses were carried out on a STA 7200 Hitachi^®^ (Fukuoka, Japan). Briefly, both freeze-dried NPs formulations were heated up to 400 °C, at a heating rate of 10 °C/min under nitrogen atmosphere (20 mL/min).

### 2.5. Production of ZeinCPX_HA NPs and Stability Studies

The production of ZeinCPX_HA NPs was performed by following the methodology described in Section 2.2, where the CPX (4 mg/mL in 0.1 N HCl) was dissolved in the zein solution in a volume ratio of 1:10 (CPX and zein). Then, the characterization of ZeinCPX_HA NPs was conducted by using the same procedures outlined in Section 2.4.

Further, the stability studies were performed on both NPs (Zein_HA NPs and ZeinCPX_HA NPs). Briefly, NPs were rehydrated, after lyophilization, by adding ultrapure water and left under static conditions for 10 min at room temperature (22 ± 2 °C) before being fully reconstituted by manual shaking. Then, the lyophilized NPs were stored at room temperature for 28 days, redispersed in ultrapure water every 7 days and characterized as described in Section 2.4. Additionally, the RDI of the ZeinCPX_HA NPs was also calculated according to Equation (1) of Section 2.3.

### 2.6. Encapsulation Efficiency and Loading Capacity

The EE and drug loading (DL) of CPX into the NPs were assessed in the supernatant after centrifuging the NPs (4000× *g*; 10 min), using a 10 kDa Amicon^®^ Ultra-2 Centrifugal Filter Device (Merck Millipore, Darmstadt, Germany). Then, the CPX was quantified by HPLC as previously described by [37]. The chromatographic analysis was performed using an UltiMate 3000 HPLC chromatography device (Thermo Scientific, Waltham, MA, USA) with a column C18 (Acclaim™ 120 Reversed-Phase Columns C18, 5 μm, 4.6 × 150 mm, Thermo Scientific, Waltham, MA, USA) at a temperature of 40 °C. The mobile phase consisted of 0.025 M ortho-phosphoric acid and acetonitrile (87:13 *v*/*v*) with a flow rate of 0.9 mL/min, and the injection volume was 20 µL. The run time cycle was completed in 20 min. CPX was detected at 278 nm with a retention time of ≈11 min. All experiments were carried out in triplicate. The standard calibration curve was obtained (y = 2.5253x − 0.707; R^2^ = 0.9993).

The EE was calculated by using the following Equation (2)
(2)$$\text{EE (\%)}=\frac{A_{1}-A_{2}}{A_{1}}\times 100$$
where A_1_ is the total amount of CPX added into NPs and A_2_ is the amount of drug in the supernatant.

The DL was determined using the following Equation (3)
(3)$$\text{DL (\%)}=\frac{\text{Weight of the drug in NPs}}{\text{Weight of the NPs}}\times 100$$

### 2.7. In Vitro Drug Release of CPX-Loaded NPs

The in vitro release of CPX from lyophilized NPs was performed in simulated tear fluid (STF), which contains 0.22 g of sodium bicarbonate, 0.68 g of sodium chloride, 0.008 g of calcium chloride dehydrate, and 0.14 g of potassium chloride, dissolved in 100 mL of distilled deionized water to mimic ocular physiological conditions [38,39]. In brief, ZeinCPX_HA NPs were immersed in STF (pH 7.4) at 600 mg/mL, and then 1 mL was collected in microcentrifuge tubes at different time points for 24 h, under continuous agitation at 37 °C. At precise time points (0, 1 h 30 min, 2 h 30 min, 4 h 30 min, 6 h 30 min, 8 h 30 min and 24 h), the corresponding microcentrifuge tube was centrifuged (4000× *g*, 10 min, 25 °C) through a 10 kDa Amicon^®^ Ultra-2 Centrifugal Filter Device (Merck Millipore, Darmstadt, Germany), and the CPX remaining in the supernatant was quantified by HPLC-UV (Thermo Scientific, Waltham, MA, USA). All experiments were performed in triplicate. Zein_HA NPs without CPX were used as a control.

Furthermore, the drug-release kinetics displayed by the NPs was also characterized by applying the zero-order Equation (4), first-order Equation (5), Peppas–Korsmeyer Equation (6) and Higuchi model Equation (7):(4)$$\text{Zero order: }D_{t}=D_{0}+k_{0}t$$
(5)$$\text{First order: log C}=\text{log}C_{0}-\frac{K_{t}}{2.303}$$
(6)$$\text{Korsmeyer-Peppas: }\frac{\mathrm{Mt}}{M\infty }=Kt^{n}$$
(7)$$\text{Higuchi: }Q_{t}=K_{H}\;t^{1/2}$$
where D_t_ is the amount of drug released at time t, D_0_ is the initial drug amount in the solution, and k_0_ is the constant release rate. C_0_ is the initial drug concentration, K is the first-order rate constant. M_t_ is the cumulative amount of drug released at time t, M∞ is the initial drug loading, K is a constant characteristic of the drug-polymer system, and *n* is the diffusion exponent, indicating the release mechanism. Q_t_ is the amount of drug dissolved in time t, and K_H_ is the Higuchi dissolution constant.

### 2.8. Biocompatibility Assay

The biocompatibility of both NPs (with and without CPX) was assessed on normal human dermal fibroblasts (NHDF) cells by the MTT assay, according to the ISO 10993-5 “*Biological evaluation of medical devices—Part 5: Tests for in vitro cytotoxicity*”. Briefly, NHDF cells were seeded in 96-wells plates (6000 cells/well) and maintained in DMEM-F12 medium supplemented with 10% FBS and 1% penicillin–streptomycin–amphotericin B at 37 °C in 5% CO_2_ humidified atmosphere. After 24 h of incubation, the culture medium was replaced by NPs solutions cryopreserved with 5% of mannitol (with and without CPX) at different concentrations (12.5, 25, 50, 100 and 125 mg/mL). Following 24 h of incubation, the medium in each cell was replaced by a mixture of 50 µL of MTT solution (5 mg/mL) and incubated for 4 h at 37 °C and 5% CO_2_. Then, the MTT solution was removed, and cells were treated with 100 µL of DMSO (0.04 N) for 30 min. Afterwards, the absorbance of the wells (*n* = 5) was determined at 570 nm using a microplate reader (Thermo Scientific Multiskan GO UV/Vis microplate spectrophotometer, Waltham, MA, USA). Cells incubated with ethanol (96%) were used as a positive control (K^+^), whereas untreated cells were used as negative control (K^−^).

## 3. Results and Discussion

### 3.1. Effect of Polymers’ Concentration on Their Properties

NPs produced with lower concentrations of HA and zein resulted in smaller NPs with low PDI (Table 1).

When zein and HA had a concentration of 5 and 2.5 mg/mL, respectively, the produced NPs had a Dh higher than 200 nm, which is unsuitable to permeate the mucus [40]. On the other hand, when zein concentration decreased to 2.5 mg/mL the resulting NPs had a lower Dh, but with a PDI over 0.5. So, the NPs produced with 1 mg/mL of HA and 2.5 mg/mL of zein, presented the most suitable Dh and PDI, being selected for subsequent analysis.

### 3.2. Selection of the Cryoprotectant

The freeze-drying process was conducted to evaluate the stability of the nanosystem for long-term storage. The lyophilization process strongly affected the Dh and PDI of Zein_HA NPs, as shown in Figure 1. As for, glucose and mannitol were used at different concentrations (5 and 10% *w*/*v*) as cryoprotectants.

As shown in Figure 1, the addition of cryoprotectant affected the NPs physicochemical properties, namely Dh, PDI and zeta potential. The use of mannitol at 5% *w*/*v* does not affect the physicochemical properties; however, the Dh of the Zein_HA NPs increased from ≈86 nm to ≈160 nm after using the mannitol at 10% *w*/*v*, whereas when used glucose as a cryoprotectant, the Dh increased to ≈589 nm and ≈236 nm at 5 and 10% *w*/*v*, respectively. These results are in accordance with the findings presented by Wang et al., where the authors also denoted an increase in the NPs after lyophilization [41]. Further, results show that NPs cryoprotected with mannitol resulted in NPs with a Dh, PDI, and zeta potential more similar to the fresh NPs. In light of these findings, the RDI values were calculated for mannitol at 5 and 10% since these cryoprotectants did not affect the Dh and PDI of Zein_HA NPs (Table 2).

The RDI obtained for mannitol at 5 and 10% was 116 ± 32 and 187 ± 72%, respectively. Taking into account the results obtained, the Zein_HA NPs produced and freeze-dried with mannitol at 5% were selected for the subsequent studies. This result is in accordance with those previously obtained for cryoprotected zein NPs [36]. Moreover, Gagliadri et al. demonstrated that the protective effect of mannitol on zein NPs could be related to its physicochemical properties, which can prevent Maillard reactions that commonly occur when subjected to freeze-drying [25]. Additionally, Feng et al. suggest using mannitol once it crystalizes around the NPs, forming a protective shell and ultimately preventing aggregation [42].

### 3.3. Characterization of NPs Incorporating CPX

Initially, the successful production of both NPs was also confirmed by FTIR analysis after the freeze-drying process (Figure 2).

The FTIR spectrum of zein is in line with previous reports, exhibiting its characteristic peaks at 3292 cm^−1^ (–OH stretching), 1643 cm^−1^ (C–O stretching and C–N stretching, amide I), and 1515 cm^−1^ (C–N stretching and N–H bending, amide II) [43,44,45]. The HA spectrum displays its characteristic peaks at 3200–3650 cm^−1^ (O–H stretching), 2900 cm^−1^ (CH_2_ vibration), 1607 cm^−1^ (carboxyl group in the glucuronic unit), and 1035 cm^−1^ (C–O–C stretching) [43,46,47]. The typical band of CPX was displayed at 1612 cm^−1^ (vibration of phenyl structure conjugated to –COOH), 1282 cm^−1^ (C–F bond stretching), 3044, and 2844 cm^−1^ (C–H stretching from the phenyl ring) [48,49,50]. In turn, mannitol FTIR spectrum presents its characteristic peaks between 3386 and 2902 cm^−1^ (O–H and C–H stretching vibration) and at 1417, 1280, and 1077 cm^−1^ [51].

The FTIR spectrum of ZeinCPX_HA NPs displays high similarity with the Zein_HA NPs spectrum, without relevant peak shifts or new peaks. According to Cacicedo et al., this can be due to the low ratio of CPX in NPs compared to the polymers and cryoprotectants, or the overlap of peaks between CPX and the other compounds [52]. Concerning the presence of both polymers in the NPs, there was a slight shift in the NPs spectra from 3292 cm^−1^ (zein) and 3200–3650 cm^1^ (HA) to 3186 cm^−1^.

As shown in Figure 3A, the TGA thermogram representing HA weight loss exhibits a water loss of about 12% up to 220 °C, in the second step occurs a weight loss of approximately of 55.2% at 220–280 °C due to the polysaccharide degradation, and at the end there is a linear weight loss of close to 10% up to 400 °C [53,54]. CPX TGA thermogram (Appendix A) presents a significant weight loss at 250–450 °C, which is in accordance with the melting point of ≈280–290 °C presented in the DSC thermogram (Appendix A).

Regarding zein, it presents two decomposition steps. The first step of about 9% weight loss occurs due to the water loss between 100–250 °C. The second step shows a weight loss of 79%, at 250–500 °C due to the degradation of peptide bounds [55]. On the DSC thermogram (Appendix A), zein endothermic peak is 115 °C. As for mannitol, TGA thermogram revealed a weight loss of higher than 90% between 250–400 °C, and regarding the DSC thermogram (Appendix A) mannitol endothermic peak registered at 170 °C [56].

The Zein_HA NPs exhibit a weight loss of −84.1% at 200–320 °C. On the other hand, ZeinCPX_HA NPs have a weight loss of 96% at 199–340 °C; this slight shift can be due the encapsulation of CPX. As shown on the DSC thermogram, Zein_HA NPs and ZeinCPX_HA NPs present an endothermic peak at 165.5 °C and 163 °C, respectively. The endothermic peak of CPX does not appear in the ZeinCPX_HA NPs, so CPX can be dispersed in amorphous state, which can enhance the solubility of the drug and consequently, their bioavailability [57,58].

Ocular drug delivery is challenging due to physiological barriers, limitations associated with conventional ocular therapy (i.e., blurred vision and frequent administration), and NPs’ properties, such as Dh, charge, and hydrophilicity. It is extremely important to overcome the challenges mentioned above [59,60,61]. The Dh of the fresh NPs suffered a significant increase upon the encapsulation of CPX from 86 ± 14 nm to 109 ± 10.2 nm, as shown in Figure 4. This increment can be due to the incorporation of the drug, which is in accordance with other authors’ findings. For instance, Nunes et al. demonstrated that the Dh of zein NPs increased from 129 ± 3 nm to 141 ± 7 nm, with the augment of resveratrol loading [20]. Ye et al. also reported an increase in the Dh of polysaccharide/zein NPs after the encapsulation of doxorubicin [44]. The Dh of ZeinCPX_HA NPs is suitable for ocular drug delivery, given that NPs with Dhs between 50 and 200 nm can permeate through the ocular mucous [40]. Furthermore, the incorporation of HA is expected to improve the cellular uptake by ophthalmic cells, as described by Apaolaza et al. for HA-coated gold NPs [62]. In addition, HA is present in the eye constitution [63,64] and described as promising mucoadhesive polymer for ocular drug delivery systems development [65,66]. Thus, this mucoadhesiveness property results on an increase of drug ocular residence time [63,67].

Moreover, PDI is another crucial characteristic in assessing the successful production of NPs, indicating the formulations’ homogeneity [68]. There was a slight increase in the PDI (from 0.21 ± 0.07 to 0.27 ± 0.03) after the loading with CPX; however, it remains below 0.3, which indicates that ZeinCPX_HA NPs are monodisperse.

The surface charge of the NPs also plays an important role in the stability and cell interaction. The ZeinCPX_HA NPs presented a zeta potential of −33 ± 4.2, which is considered suitable for achieving colloidal stability [69]. Since ocular mucosa has an anionic character, HA can bind to the negatively charged mucin in the corneal and conjunctival epithelium [27].

Next, the stability of lyophilized NPs with 5% mannitol was evaluated for 28 days, as shown in Figure 5.

During the 28 days, the Dh of ZeinCPX_HA NPs remained stable, and the PDI of both NPs did not suffer significant differences.

Furthermore, during the 28 days, there were no changes in the zeta potential of Zein_HA NPs and ZeinCPX_HA NPs, which means that the nano-formulations are chemically stable [51]. Besides, the obtained zeta potential values are close to –30 mV, so according to the literature, nano-formulations with zeta potential values more than +30 mV and less than −30 mV are considered stable [70].

Moreover, the RDI values were also calculated, corroborating the efficiency of the NPs reconstitution, given that the RDI value of the ZeinCPX_HA NPs was 92 ± 13%.

### 3.4. Encapsulation Efficiency and Drug Loading

In the clinic, the concentration of ciprofloxacin on eye drops used for topical treatment of ocular infections is around 3000 mg/L. This treatment implies the administration of 1–2 drops every 15–30 min (initially) in acute infections and 1–2 drops application 6 times per day, or more, in severe conditions [71,72,73]. On the other hand, the incorporation of ciprofloxacin into polymeric drug delivery system will allow to decrease the drug concentration to be used, and will assure a controlled drug release and a prolonged therapeutic effect, as has been demonstrated by Günday et al. for ciprofloxacin-loaded poly(ᴅʟ-lactide-*co*-glycolide) NPs with a CPX concentration of 20 µg/mL [74]. This CPX concentration was incorporated into ZeinCPX_HA NPs, with an EE of 69 ± 5% and a loading content of 3.7 ± 1.4%, indicating that CPX was successfully encapsulated through the FNP technique. To the best of our knowledge, this is the first report on encapsulating CPX into zein nanosystems. Fu et al. (2009) obtained an EE between 4.97% and 8.29% and a drug loading ranging from 0.87 to 2.41%, when encapsulating CPX (2 to 5 mg/mL) into zein (8 to 20 mg/mL) microspheres [75]. Furthermore, the EE of ZeinCPX_HA NPs is in line with other studies with polymeric NPs, which reported an EE of CPX ranging between 51.8 and 82.7% [74,76,77]. For CPX-loaded PLGA and PEG NPs produced by nanoprecipitation, there was an EE of 25 ± 9% and a loading rate of 3.6 ± 1.3% [78]. Additionally, Xu et al., reported low EE of CPX in PLA-DEX NPs (1.1 ± 0.04% (CPX at 4 mg/mL) and a mass loading of 8.45 ± 0.31%) in PLGA-PEG NPs the EE was 1.35 ± 0.07% with a loading of 10.79 ± 0.59% [79]. Furthermore, the encapsulation of other lipophilic compounds into zein NPs is comprehended between 47.80 and 98% [18,24,80,81,82,83].

### 3.5. CPX Release Profile from ZeinCPX_HA NPs

After assessing the drug encapsulation and loading, the in vitro release of CPX from lyophilized ZeinCPX_HA NPs was studied in STF. As shown in Figure 6, the release profile of CPX displays an initial burst release in the first 1 h and 30 min of incubation, followed by a prolonged release for at least 24 h. This initial burst release can be due to the hydrophilic nature of HA, which rapidly dissolves in STF and releases the drug [84]. Then, the decrease of the released drug can be attributed to the CPX entrapment in the hydrophobic core (zein) of the ZeinCPX_HA NPs [22,85]. This is in accordance with previous reports on improved ability of nanosystems for ocular drug delivery; however, a formulation providing a sustained release for CPX is not available as commercial dosage form. This was already discussed in a comparative analysis of commercial topical dosage forms (e.g., 0.3% CPX-hydrochloride Ciloxan^®^ drop, Alcon Laboratories Inc., Fort Worth, TX, USA) with drug release carriers, highlighting that: (i) the MIC of CPX incorporated into liposomal formulations was lower, and with an increased antibacterial effect; (ii) the ocular bioavailability of CPX was improved in comparison with Ciloxan^®^ ophthalmic drops; and (iii) the drug residence time on corneal surface was improved by the surface properties of the nanosystem [10].

Furthermore, applying mathematical models also assessed the kinetics of CPX release. The release of CPX by the NPs follows Korsmeyer–Peppas kinetic model (R^2^ = 0.9505 and *n* < 0.43), since this model presents the correlation coefficient (R^2^) closer to 1 (Table 3). The coefficient value was acquired by plotting the log of the percentage of drug release versus log time [84]. The Korsmeyer–Peppas results also indicated that the CPX release occurs due to a Fickian diffusion process (*n* ≤ 0.45), resulting from the swelling of the polymeric matrix [19,86,87].

### 3.6. Characterization of NPs’ Biological Properties

The biocompatibility of both NPs (with and without CPX) was confirmed on NHDF cells (Figure 7). NHDF cells were used as a cell model, given that these cells are constituents of the subepithelial layer of the conjunctiva [88]. Additionally, these cells recruit neutrophils and monocytes at an early stage of infection regardless of the type of microorganism [89].

Our results demonstrated that the NPs are biocompatible, which is expected considering the reported results in other studies for zein and HA NPs. Our results demonstrated that the NPs are biocompatible, which was expected considering the reported results in other studies for zein and HA NPs, as well as their approval by the FDA and wide application in the pharmaceutical field [20,21,44,46]. Furthermore, HA is widely employed in commercial dosage forms for ocular disorders treatment, such as Healon^®^, Amvisc^®^, Provisc^®^ and AMO Vitrax^®^ [26]. These results highlighted that the ZeinCPX_HA NPs could act as a drug delivery system. However, in further studies assuring the translation potential of this formulation, additional assays (preclinical and clinical evaluation) would be extremely important to assure a complete characterization of this new therapeutic system for clinics.

## 4. Conclusions

In this study, zein and hyaluronic acid were used for the first time to produce nanoparticles by the flash nanoprecipitation technique aimed to encapsulate ciprofloxacin to work as a topical drug delivery system to treat ocular mucosa disorders, such as bacterial conjunctivitis. The obtained results demonstrate that the nanoparticles cryoprotected with 5% mannitol were stable upon the freeze-drying process, as corroborated by the RDI, size, PDI, and zeta potential values.

Overall, our results support the application of ZeinCPX_HA NPs as a possible alternative to the current antibacterial topical dosage forms available on the market to treat conjunctivitis. In the future, the investigation of antibacterial and ex vivo assays can be achieved to evaluate further the potential of these nanosystems for translation to the clinical management of bacterial conjunctivitis.

## Figures and Tables

**Figure 1 pharmaceutics-14-01557-f001:**
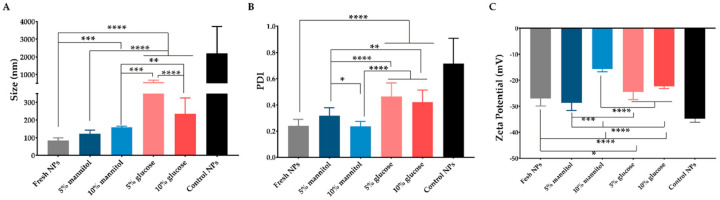
Evaluation of the physicochemical properties of Zein_HA NPs with and without cryoprotectants. Data are presented as mean ± standard deviation (*n* > 3), * < *p* = 0.05, ** < *p* = 0.01, *** < *p* = 0.001, **** *p* < 0.0001.

**Figure 2 pharmaceutics-14-01557-f002:**
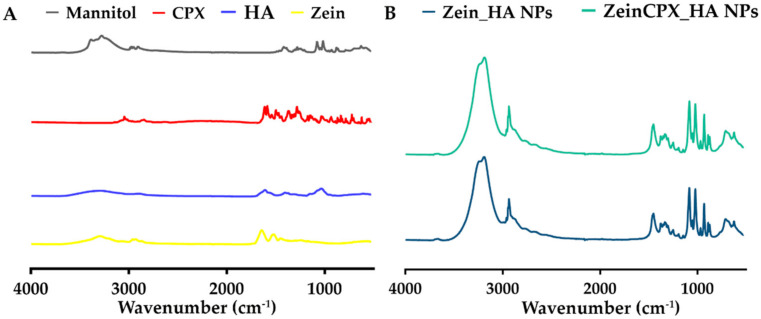
FTIR spectra of the NPs and raw materials: (**A**) spectra of the raw materials (zein, HA, CPX, and mannitol) used for the production and lyophilization of the NPs; (**B**) spectra of the Zein_HA NPs and ZeinCPX_HA NPs.

**Figure 3 pharmaceutics-14-01557-f003:**
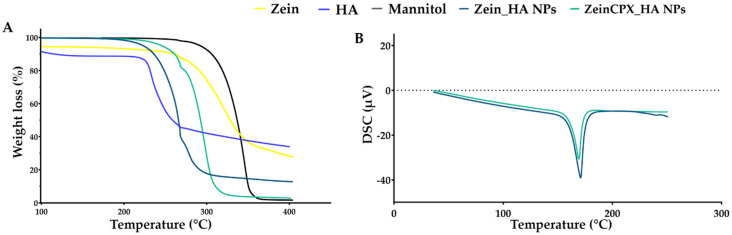
TGA (**A**) and DSC (**B**) analysis of the raw materials (zein, HA and mannitol), Zein_HA NPs and ZeinCPX_HA NPs.

**Figure 4 pharmaceutics-14-01557-f004:**
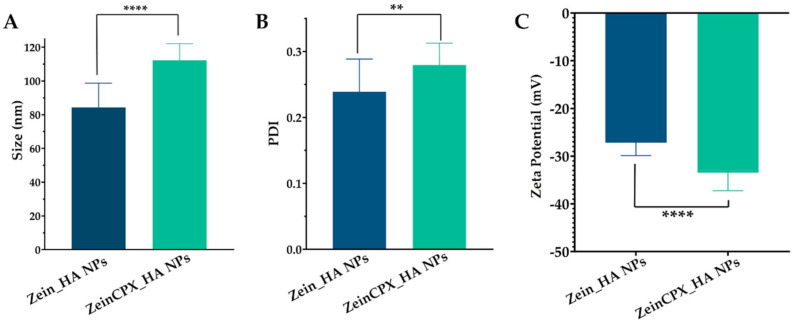
Evaluation of morphological properties of Zein_HA NPs and ZeinCPX_HA NPs: determination of the hydrodynamic diameter values (**A**), PDI (**B**), and zeta potential (**C**). Data are presented as mean ± standard deviation, *n* = 5, ** < *p* = 0.01, *****p* < 0.0001.

**Figure 5 pharmaceutics-14-01557-f005:**
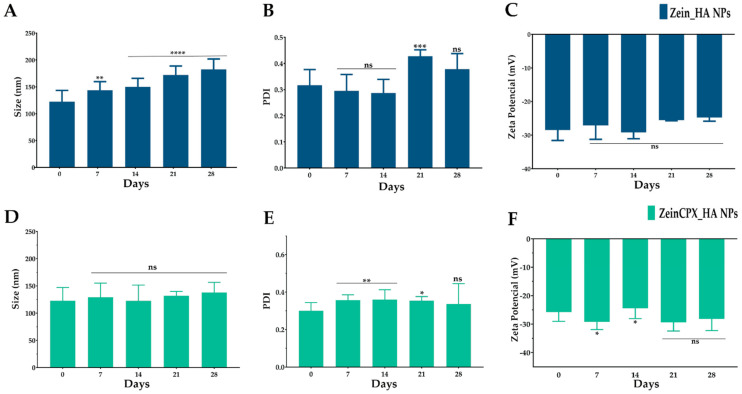
Stability of Zein_HA NPs (**A**–**C**) and ZeinCPX_HA NPs (**D**–**F**), after lyophilization, for 28 days. Data are presented as mean ± standard deviation, *n* = 5, ns *p* > 0.05, * < *p* = 0.05, ** < *p* = 0.01, *** *p* < 0.001, ****< *p* = 0.0001, values marked with asterisks are statistically different from Day 0 (immediate re-hydration of the NPs upon lyophilization).

**Figure 6 pharmaceutics-14-01557-f006:**
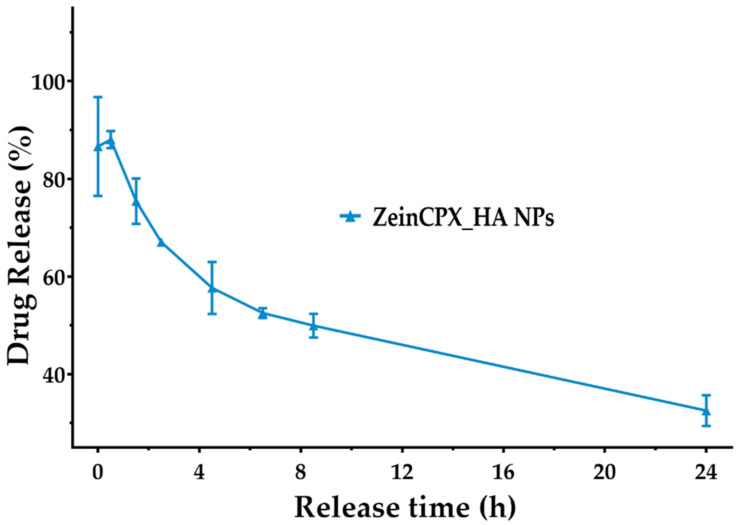
Release profile of ZeinCPX_HA NPs in STF at 37 °C, pH = 7.4 for 24 h (non-cumulative). Results are presented as mean ± SEM (*n* = 3).

**Figure 7 pharmaceutics-14-01557-f007:**
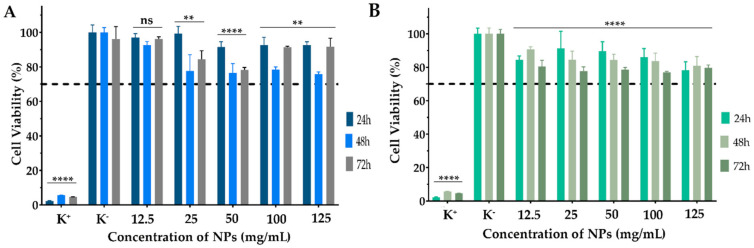
Characterization of the biocompatibility of (**A**) Zein_HA NPs cryoprotected with 5% mannitol and (**B**) ZeinCPX_HA cryoprotected with 5% mannitol NPs in contact with NHDF cells during 24 h, 48 h and 72 h. Data are presented as mean ± SEM, *n* = 5, ns—not statistically significant, ** < *p* = 0.001, **** < *p* = 0.0001, values marked with asterisks are statistically different from the mean of K^−^.

**Table 1 pharmaceutics-14-01557-t001:** Dh and PDI of the NPs with different polymer concentrations. Data are presented as mean ± SD (*n* = 3).

Polymers	Dh (nm)	PDI
HA (2.5 mg/mL) + Zein (2.5 mg/mL)	146.1 ± 7.4	0.523 ± 0.019
HA (2.5 mg/mL) + Zein (5 mg/mL)	230.3 ± 15.1	0.853 ± 0.056
HA (1 mg/mL) + Zein (2.5 mg/mL)	86 ± 14	0.21 ± 0.07

**Table 2 pharmaceutics-14-01557-t002:** RDI of the Zein_HA NPs after lyophilization with different cryoprotectants. Data are presented as mean ± standard deviation (*n* = 3).

Cryoprotectant	RDI (%)
Mannitol 5%	116 ± 32
Mannitol 10%	187 ± 72

**Table 3 pharmaceutics-14-01557-t003:** Regression coefficients of mathematical models fitted to the release of CPX from ZeinCPX_HA NPs.

Mathematical Model	R^2^	*n*
Zero order	0.7657	-
First order	0.5791	-
Higuchi	0.9269	-
Korsmeye–Peppas	0.9505	0.2635

## Data Availability

No new data were created or analyzed in this study. Data sharing is not applicable to this article.

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
