# Peer review of "Ciprofloxacin-Loaded Zein/Hyaluronic Acid Nanoparticles for Ocular Mucosa Delivery"

_pharmaceutics, 2022, doi:10.3390/pharmaceutics14081557_

Round 1

Reviewer 1 Report

Dear Author,

I would like to congratulate you on your well-designed research study. The research article entitled “Ciprofloxacin-Loaded Zein/Hyaluronic Acid Nanoparticles for Ocular Mucosa Delivery” has been intensively reviewed and evaluated. After this evaluation process hereby I would like to share some major and minor revisions.

Hereby, I would like to present my suggestions and revisions.

Revision_1 (minor): (line 38) Please check and revise the full name of the bacterium “…Pneumoniae, Haemophilus influenza…).

Revision_2 (minor): (line 116) please indicate your dispersion volume or mass that was freeze-dried.

Revision_3 (major): (line 142, total section), How could you determine/decide the active ingredient concentration, is there a pharmacological basis (reference eye drop or bioactive concentration) or is it the optimal concentration that could be loaded to the nanoparticles. Please explain and insert the explanation into the text.

Revision_4 (minor): (line 158) “…a 10 k Amicon® Ultra-2…” please revise the “k” as kDa if it indicated molecular weight. And follow this revision during the text.

Revision_5 (major): (line 180, total section), why did you choose specifically these two types of kinetic models, please explain in the text. If there is a specific situation please support with the literature. There is a need to show first order and zero order kinetics. Please insert the related equations and compare the previously presented kinetic models.

Revision_6 (minor): (line 214) “…were treated with 100uL of DMSO…” please check your symbol. Does it indicate micro? If so please revise.

Revision_7 (minor): (line 345) “…For CPX-loaded PLAG and PEG NPs produced…” please check the abbreviation, it should be PLGA.

Revision_8 (minor): (line 355) “…in the first 1h30 of incubation…” please clarify the time. Could be 1 h and 30 min or 1.5 h.

Revision_9 (major): As a general comment for the total nanoparticle characterization, there is a need to illustrate the crystal structure of the nanocarriers. Thus, SEM, TEM, XRD, DSC, TGA, or any other method could be conducted. At least one of them should be applied (DSC is the most preferred one) and discussed.

Revision_10 (major): In nanoparticle characterization, only DLS data is insufficient for a Q1/Q2 level study. Therefore, the DLS data should be supported with other techniques such as LD or visual methods (SEM or TEM).

Best wishes.

Author Response

Dear Author,

I would like to congratulate you on your well-designed research study. The research article entitled “Ciprofloxacin-Loaded Zein/Hyaluronic Acid Nanoparticles for Ocular Mucosa Delivery” has been intensively reviewed and evaluated. After this evaluation process hereby I would like to share some major and minor revisions.

Hereby, I would like to present my suggestions and revisions.

  1. (line 38) Please check and revise the full name of the bacterium “…Pneumoniae, Haemophilus influenza…).

Thank you. The species identification has been revised and corrected (please see line 38, on page 1).

  1. (line 116) please indicate your dispersion volume or mass that was freeze-dried.

Thanks for the suggestion. The volume of the nanoparticles’ formulation (1 mL) is indicated in line 116, page 3.

  1. (line 142, total section), How could you determine/decide the active ingredient concentration, is there a pharmacological basis (reference eye drop or bioactive concentration) or is it the optimal concentration that could be loaded to the nanoparticles. Please explain and insert the explanation into the text.

The authors thank the reviewer for bringing this topic. This issue has been introduced in the revised manuscript (please see page 10, lines 384-393). “In the clinic the concentration of ciprofloxacin on eye drops used for topical treatment of ocular infections is around 3000 mg/l. This treatment implies the administration of 1–2 drops every 15–30 min (initially) in acute infections and 1–2 drops application 6 times per day, or more, in severe conditions [70-72]. On the other hand, the incorporation of ciprof-loxacin into polymeric drug delivery system will allow to decrease the drug concentration to be used, and assure a controlled drug release and a prolonged therapeutic effect, as has been demonstrated by Günday et al., for ciprofloxacin-loaded poly(DL-lactide-co-glycolide) NPs with a CPX concentration of 20 µg/mL [73]. This CPX concentration was incorporated into ZeinCPX_HA NPs, with an EE of 69 ± 5 % and a loading content of 3.7 ± 1.4 %, indicating that CPX was successfully encapsulated through the FNP technique.

  1. “…a 10 k Amicon® Ultra-2…” please revise the “k” as kDa if it indicated molecular weight. And follow this revision during the text.

The correction was performed; please check line 166 and line 196 on page 4.

  1. (line 180, total section), why did you choose specifically these two types of kinetic models, please explain in the text. If there is a specific situation please support with the literature. There is a need to show first order and zero order kinetics. Please insert the related equations and compare the previously presented kinetic models.

The authors appreciate this valuable suggestion. The obtained release data were fitted using zero-order, first-order, Higuchi and Korsmeyer-Peppas models. Afterwards, the kinetics that showed the best correlation coefficient (R2) was introduced in the manuscript, as well as the zero and first-order kinetics R2 values and equations (please see lines 200 to 215 on page 5 and Table 3, page 11).

  1. (line 214) “…were treated with 100uL of DMSO…” please check your symbol. Does it indicate micro? If so please revise.
  2. …For CPX-loaded PLAG and PEG NPs produced…” please check the abbreviation, it should be PLGA.
  3. “…in the first 1h30 of incubation…” please clarify the time. Could be 1 h and 30 min or 1.5 h.

Thank you. The typos were corrected.

  1. As a general comment for the total nanoparticle characterization, there is a need to illustrate the crystal structure of the nanocarriers. Thus, SEM, TEM, XRD, DSC, TGA or any other method could be conducted. At least one of them should be applied (DSC is the most preferred one) and discussed

The authors thank the reviewer for the suggestion. As suggested, DSC and TGA analysis were conducted and discussed. So, these data were introduced and discussed in materials and methods section (please see lines 144-148 on page 3) and in results and discussion section (please see lines 306-330, page 7 and 8; and Figures S1 and S2).

  1. In nanoparticle characterization, only DLS data is insufficient for a Q1/Q2 level study. Therefore, the DLS data should be supported with other techniques such as LD or visual methods (SEM or TEM).

The authors thank the reviewer for the important comment. Indeed, SEM/TEM analysis are usually performed to characterize the morphology of the nanoparticles. However, the DLS analysis also evaluated such property regarding the quantitative feature, and it is most technique used for the characterization of the nanoparticles in pharmaceutical technologies (Gioria et al., 2018).

SEM analysis of the nanosystems were performed, however it is possible to verify that the equipment has not resolution enough to acquire perceptible images (Figure 1). In turn, the TEM analysis was also required but the equipments and technicians are quite busy, being not possible the realization of this analysis within review time.  Despite of, the images showed the spherical shape of the nanosystems, corroborating the DLS analysis. Furthermore, the hydrodynamic diameter values ranging from 50- 400 nm are considered versatile for ocular delivery as they have the ability to overcome physiological barriers and to direct the drug to specific cells, either by passive or ligand mediated targeting mechanisms (Almeida, Amaral, Lobão, Silva, & Loboa, 2014).

Figure 1. Acquired images in a TM3030 Plus TableTop Hitachi® (Fukuoka, Japan). Please see attached document.

References

Almeida, H., Amaral, M. H., Lobão, P., Silva, A. C., & Loboa, J. M. (2014). Applications of polymeric and lipid nanoparticles in ophthalmic pharmaceutical formulations: present and future considerations. J Pharm Pharm Sci, 17(3), 278-293. doi:10.18433/j3dp43

Gioria, S., Caputo, F., Urbán, P., Maguire, C. M., Bremer-Hoffmann, S., Prina-Mello, A., . . . Mehn, D. (2018). Are existing standard methods suitable for the evaluation of nanomedicines: some case studies. Nanomedicine (Lond), 13(5), 539-554. doi:10.2217/nnm-2017-0338

Reviewer 2 Report

1.       What strength/dose of drug used?

2.       Zein was obtained from Acros Organics, Is it pharmaceutical grade?

3.       Line 151: Room Temp. Need specific temperature. Why researcher not followed ICH guideline?

4.       Is formulation mucoadhesive? If not how it can retain at site for 24 hours.

5.       Is ocular irritation/safety study done? Any report of non irritancy of Zein HA NPs.

6.       Is any marketed SR formulation of drug available for ocular? If yes why not compared. If no, what may be reasons?

7.       Shorten the conclusion, look like a summary.

Author Response

  1. What strength/dose of drug used?

The authors thank the reviewer for the question. This issue has been introduced in the revised manuscript (please see page 10, lines 384-393). “In the clinic the concentration of ciprofloxacin on eye drops used for topical treatment of ocular infections is around 3000 mg/l. This treatment implies the administration of 1–2 drops every 15–30 min (initially) in acute infections and 1–2 drops application 6 times per day, or more, in severe conditions [70-72]. On the other hand, the incorporation of ciprof-loxacin into polymeric drug delivery system will allow to decrease the drug concentration to be used, and assure a controlled drug release and a prolonged therapeutic effect, as has been demonstrated by Günday et al., for ciprofloxacin-loaded poly(DL-lactide-co-glycolide) NPs with a CPX concentration of 20 µg/mL [73]. This CPX concentration was incorporated into ZeinCPX_HA NPs, with an EE of 69 ± 5 % and a loading content of 3.7 ± 1.4 %, indicating that CPX was successfully encapsulated through the FNP technique.

  1. Zein was obtained from Acros Organics, Is it pharmaceutical grade?

The authors thank the reviewer for the question. This information was added to the introduction section; please see lines 63-64 and line 95, page 2. “Herein, ZeinCPX_HA NPs were prepared by the FNP technique. Zein is a water-insoluble protein extracted from corn and is generally considered safe (GRAS) by the FDA [20]. In fact, zein has been widely explored in biomedical applications, namely in the field of pharmaceuticals due to its physicochemical and biological properties [21].”

  1. Line 151: Room Temp. Need specific temperature, Why researcher not followed ICH guideline.

The authors thank the reviewer for the observation. The specific temperature was 22 ±2°C.  The correction was performed; please check line 157, page 4. In this work we performed the stability assays for 28 days, however in further studies the critical quality attributes of the nanosystems under long-term storage conditions will be monitored following ICH guidelines.

  1. Is formulation mucoadhesive? If not how can retain at site for 24 hours.

The authors thank the reviewer for bringing this important topic. This information was added in the revised manuscript; please see line 345-348, on page 8. “In addition, HA is present in the eye constitution [62,63] and described as promising mucoadhesive polymer for ocular drug delivery systems development [64,65]. So, this mucoadhesiveness property results on an increase of drug ocular residence time [62,66].”

  1. Is ocular irritation/safety study done. Any report of non irritancy of Zein HA NPs.

Thank you for the important observation. This was discussed on lines 449-457, on page 12). “Our results demonstrated that the NPS are biocompatible, which is expected considering the reported results in other studies for zein and HA NPs and once these polymers are approved by FDA and are widely applied in the pharmaceutical field [20,44,46,89]. Furthermore, HA is widely employed in commercial dosage forms for ocular disorders treatment, such as Healon®, Amvisc®, Provisc® and AMO Vitrax® [26].  These results highlighted that the Zein CPX_HA NPs could act as a drug delivery system. However, in further studies assuring the translation potential of this formulation, additional assays (preclinical and clinical evaluation) would be extremely important to assure a complete characterization of this new therapeutic system for clinics.”

  1. Is any marketed SR formulation of drug available for ocular? If yes why not compared. If no, what may be reasons?

Thank you for the question. This was discussed in the revised version of the manuscript (please see lines 413-422, pages 10-12). “This is in accordance with previous reports on improved ability of nanosystems for ocular drug delivery; however, a formulation providing a sustained release for CPX is not available as commercial dosage form. This was already discussed in a comparative analysis of commercial topical dosage forms (e.g. 0.3% CPX-hydrochloride (Ciloxan® drop, Alcon Laboratories Inc., Fort Worth, TX, USA) with drug release carriers, highlighting that: i) the MIC of CPX incorporated into liposomal formulations was lower, and with an increased antibacterial effects; ii) the ocular bioavailability of CPX was improved in comparison with Ciloxan® ophthalmic drops; and iii) the drug residence time on corneal surface was improved by the surface properties of the nanosystem [10].

  1. Shorten the conclusion, look like a summary.

We would like to thank the reviewer for the suggestion. The conclusion was revised (please see lines 459-476, page 12).

Round 2

Reviewer 1 Report

Dear author,

I am glad to receive your revised manuscript. The majority of revisions were accomplished and evaluated as sufficient. In DLS analysis the morphology (particle size actually) has been found by using scattering equations, thus it needs to be supported by visual methods to prove the real dimensions. This is why it was requested in the previous revision series. However, considering the availability and suitability of the devices (in light of the data you have provided), I exclude this revision request. In summary, I would like to inform you the revised manuscript could be accepted after the editor's decision.

Best regards.